

# Single-cell data revealed the function of natural killer cells and macrophage cells in chemotherapy tolerance in acute myeloid leukemia

Jing Gao, Xueqian Yan, Dan Fan and Yuanchun Li

Department of Hematology, The Second Affiliated Hospital of Air Force Medical University, Xi'an, China

## ABSTRACT

**Background**. Acute myeloid leukemia (AML) is highly prevalent and heterogeneous among adult acute leukemias. Current chemotherapeutic approaches for AML often face the challenge of drug resistance, and AML immune cells play an important role in the regulation of AML drug resistance. Thus, it is of key significance to explore the regulatory mechanisms of immune cells in AML to alleviate chemotherapy resistance in AML.

**Methods**. Based on AML single-cell transcriptomic data, this study revealed the differences in the expression of immune cell subpopulations and marker genes in AML patients in the complete remission group (CR) compared to AML patients in the non-complete remission group (non-CR) after chemotherapy. Functional enrichment by clusterprofiler revealed the regulatory functions of differentially expressed genes (DEGs) in AML. AUCell enrichment scores were used to assess the immunoregulatory functions of immune cells. Pseudotime analysis was used to construct immune cell differentiation trajectories. CellChat was used for cellular communication analysis to elucidate the interactions between immune cells. Survival analysis with the R package "survival" revealed the role of immune cell marker genes on AML prognosis. Finally, the wound healing and trans-well assay were performed.

**Results**. Single-cell clustering analysis revealed that NK/T cells and macrophage cells subpopulations were significantly higher in non-CR AML patients than in CR AML. AUCell enrichment analysis revealed that FCAR+ and FCGR3A+ macrophages were significantly more active in the non-CR group and correlated with processes regulating cellular energy metabolism and immune cell activity. Differentially expressed NK cell marker genes between CR and non-CR groups mainly included *HBA1*, *S100A8*, and *S100A9*, which were associated with cancer drug resistance regulation, these marker genes of (*FCAR*, *FCGR3A*, *PREX1*, *S100A8* and *S100A9*) were upregulated in human chronic myeloid leukemia cells (HAP1) and silencing of *S100A8* affected migration and invasion of HAP1 cells. In particular, the differentiation pathways of macrophages and NK cells in non-CR differed from those of patients in the CR group. Cellular communication analyses showed that ligand-receptor pairs between NK cells and macrophage cells mainly included HLA-E-KLRK1, HLA-E-KLRC1, HLA-E-CD94:NKG2A, CLEC2B-KLRB1. In addition, LGALS9-CD45, CCL3L1- CCR1, CCL3-CCR1 between these two immune cells mainly regulate secreted signaling to mediate AML progression. Marker genes in NK/T cells and macrophage cells were significantly associated with AML prognosis.

Corresponding author
Yuanchun Li, lyc_0602@126.com

**Conclusion**. This study reveals the potential role of NK cells and macrophages in AML chemoresistance through the analysis of single-cell RNA sequencing data. This provides new ideas and insights into the key mechanisms of immune cells in AML treatment.

## INTRODUCTION

Leukemia is a malignant clonal disease originating in the hematopoietic system characterized by abnormal proliferation and impaired differentiation of leukemic cells, resulting in suppressed production of normal blood cells (*Whiteley et al., 2021*; *Park et al., 2021*). Acute myeloid leukemia (AML) is a highly heterogeneous malignant hematologic disease caused by uncontrolled clonal proliferation of hematopoietic stem cells (*Hasserjian et al., 2020*). AML has the highest incidence of acute leukemia in adults, is highly heterogeneous, progresses rapidly, and has an extremely poor prognosis (*Liu, 2021*). Currently, the treatment of AML is based on chemotherapy, in which the "7+3" chemotherapy regimen involves three consecutive days of treatment with seven drugs (including cyclophosphamide, adriamycin, vincristine, methotrexate, dexamethasone, levamonase, and rituximab) followed by four days of rest before the next round of treatment (*Estey, 2016*; *Shimony, Stahl & Stone, 2023*). This regimen has achieved remarkable results in the treatment of AML and is considered one of the standard chemotherapy regimens for AML. However, due to the heterogeneity and drug resistance of leukemia cells, as well as individual differences of patients, some patients still cannot achieve complete remission after "7+3" chemotherapy, and the tumors are not completely eliminated, leading to recurrence and metastasis (*Othus et al., 2023*). Therefore, new perspectives are needed to find therapeutic options to alleviate chemotherapy tolerance in leukemia.

Immune escape of AML cells in the microenvironment of myeloid immunosuppression is an important factor in disease recurrence and refractoriness, and adjuvant therapies targeting immune cells have a potential role in alleviating AML chemotherapy resistance (*Daver et al., 2020*; *Ladikou et al., 2020*). For example, major histocompatibility antigen (MHC) class I and II molecules on the surface of tumor cells have been identified in a variety of human cancers, especially recurrent AML, by mediating T cell recognition and killing of tumor cells, suggesting that AML cells can evade the body's immune surveillance by decreasing their immunogenicity (*Del Campo et al., 2012*; *Stolzel et al., 2012*). In AML, aberrant activation of PI3K/AKT and NF-κB pathways mediates chemoresistance in this disease, and the activation of these pathways is closely related to the differentiation of T cells and cell cycle regulation (*Evangelisti et al., 2020*; *Kodous, Balaiah & Ramanathan, 2023*). Expression of the T-cell surface co-stimulatory molecule, CD28, is involved in the activation of the PI3K/AKT pathway by enhancing the signaling of the T-cell receptor, which affects chemoresistance in AML (*Parry et al., 2005*). The NF-κB pathway regulated by T cells participates in the drug resistance process by up-regulating the expression of genes

related to multi-drug resistance in leukemia cells, such as p-protein and anti-apoptotic protein Bcl-2 (*Song et al., 2019*). In addition, it has been shown that regulatory T cells (Tregs) are significantly increased in number in the peripheral blood and bone marrow of AML patients, and restoration of normal immune system function by inhibiting Tregs cells has been demonstrated in AML mice as an important means of fighting leukemia progression (*Zhou et al., 2009*; *Shenghui et al., 2011*). Thus, the modulatory effect of Tregs on chemoresistance of cancer cells is mainly due to their suppression of the immune response to tumor antigens, which reduces the killing effect of other immune cells on leukemia cells by releasing immunosuppressive factors such as TGF-β and IL-10 (*Yan, Zhang & Sun, 2019*). At the same time, the cells also delay the recovery of the hematopoietic system by inhibiting the regeneration and differentiation of hematopoietic stem cells, which in turn enhances the formation of chemotherapeutic toxicity and leukemia cell resistance (*Riether, 2022*; *Fang et al., 2023*). In conclusion, it is of great significance to explore the regulatory mechanisms of immune cells in leukemia samples to alleviate chemotherapy resistance in leukemia patients in the context of chemotherapy tolerance.

Lately, single-cell RNA sequencing (scRNA-seq) has emerged as a groundbreaking method for examining the transcriptomic features of various cell types (*Zulibiya et al., 2023*; *Chen et al., 2020*). This approach utilizes advanced next-generation sequencing technologies to delineate the overall gene expression patterns of individual cells, thereby enabling the exploration of previously obscured heterogeneity within cellular populations (*Liang et al., 2021*). In this study, we analyzed bone marrow single-cell sequencing data from patients with complete remission (CR) and incomplete remission (non-CR) of AML after chemotherapy to reveal the tangible regulatory roles and modulatory functions of immune cells in AML chemotherapy resistance. We also revealed the differentiation process and ligand–receptor interaction relationship of chemotherapy-resistant cell lines through the proposed time-series joint cell communication analysis. Finally, the regulatory role of immune cell subpopulations and their key marker genes in AML prognosis revealed in this study was elucidated by survival analysis. The revelation of the molecular mechanisms of immune cells in AML in this study is expected to target and improve the clinical treatment strategy of AML and provide guidance for alleviating chemotherapy resistance in AML.

## MATERIALS AND METHODS

### Single-cell RNA-seq data acquisition and processing

The scRNA-seq dataset GSE198681 of AML and its control samples was downloaded from the GEO official website (https://www.ncbi.nlm.nih.gov/geo/), which contains 3 complete remission (CR) samples of AML, and two non-complete remission (non-CR) samples of AML. The subsequent AML sample data for survival analysis in this study, the FPKM dataset of TCGA-AML, was obtained from the TCGA database (https://www.cancer.gov/ccg/research/genome-sequencing/tcga) and included 151 patients. For scRNA-seq, we used the Read10X function of the Seurat package to read down the data, retaining cells with mitochondria in the 10% and gene counts between 200–6,000 (*Stuart et al., 2019*). After PCA downscaling, we used the harmony package to remove batch effects

between samples (max.iter.harmony =20, lambda =0.5) (*Korsunsky et al., 2019*). Next, we performed UMAP downscaling based on the first 30 principal components, and finally clustered the cells into groups using the FindNeighbors and FindClusters functions.

## Screening for differentially expressed genes among cellular subpopulations

To explore the heterogeneity of gene expression patterns among cell subpopulations, we used the FindAllMarkers function to calculate the highly expressed genes in each cell subpopulation (only.pos = T, min.pct = 0.25, logfc.threshold = 0.25), followed by differentially expressed genes among subgroups using FindMarkers (*Song et al., 2023*), with $p < 0.05$, $|logFC| > 1$ as the screening criteria. In addition, for the calculation of the average gene expression values, we used the log-normalization method and transformed the raw expression values (UMI counts) for each gene into standardized log expression values (log1p transformed values). The calculation formula is as follows: Normalized expression = log2 (UMI counts per gene +1). The average expression value for each cell type is the mean of the gene expression values across all cells within that type.

## Functional enrichment analysis

We performed functional enrichment of DEGs using the clusterprofiler package (set parameters: keyType = "SYMBOL", pvalueCutoff = 0.05, qvalueCutoff = 0.1). Similarity between the differential gene functional enrichment analysis and KEGG enrichment analysis between subgroups was calculated using the simplifyEnrichment package (*Gu & Hubschmann, 2023*).

## Pseudotime analysis

We used Monocle for pseudotime analysis and the FindMarkers function to construct the set of differentially expressed genes between groups ($|log2FC|>0.25$, min.pct =0.25). The cds object was constructed using newCellDataSet, low-quality cells were filtered, and the data were downscaled using the DDRTree algorithm with the reduceDimension function. The cells were sorted using the orderCells function and plotted using the plot_cell_trajectory function, plot_genes_in_pseudotime to plot trajectories and scatter plots of genes as a function of pseudotime, respectively.

## Cellular communication analysis

To explore ligand–receptor interactions among immune cell populations, we used the CellChat software package (*Jin et al., 2021*) to calculate the probability of ligand–receptor interactions occurring between cell subpopulations and to classify interaction types into Secreted Signaling and CellCell contact categories.

## AUCell score of immune cell subpopulations for gene sets

We used the org.Hs.eg.db and GO.db packages to obtain the gene sets in the GO term about the gene sets in macrophages and NK/T cells, and calculated the gene set enrichment scores in the intracellular GO term using the AUCell package.
**Table 1  The specific primers for qPCR.**

| Genes | Forward (5′–3′) | Reverse (5′–3′) |
|---|---|---|
| FCAR | AAATACAAAAAAATTAGCCAGGCG | AAAATCAAGCTTCCATTTCCAACC |
| FCGR3A | CATTCTGGCTTTGAGGCTCCC | GTCTGGCACCTGTACTCTCCACTG |
| PREX1 | GTGAGGCTTAAATGAGATACTATATTGCT | AGGCGCTCAGTCTCACAAGTTAA |
| S100A8 | ATAGCCCATCTTACACACTGCTGC | CTCGCAGGTAATGGAGTAGTTTGTT |
| S100A9 | AACACCTGCTATTTGTCGGGC | TGAGTGCTGTGCAGGTGCTC |
| $\beta$-actin | CTCCATCCTGGCCTCGCTGT | GCTGTCACCTTCACCGTTCC |

## Cell line and quantitative PCR (qPCR) assay

Human bone marrow stromal cells (HS-5) and human chronic myeloid leukemia cells (HAP1) were obtained from the Wanwu Biotech Co. (Hefei, China). The HS-5 cells were cultured by the high glucose Dulbecco's Modified Eagle's medium (DEME), and the HAP1 cells were cultured by the complete medium (GIBCO, USA). The medium was added with 10% fetal bovine serum (FBS) and 1% antibiotics of streptomycin-penicillin and incubated the cells at 37 °C with 5% $CO_2$ atmosphere. Then, we obtained the total RNA of cells by using the Trizol reagent (Sigma-Aldrich, St. Louis, MO, USA) and the cDNA by using the Reverse Transcription Kit (Vazyme, Nanjing, China), and qPCR was conducted by the SYBR Green Master Mix (Vazyme) according to specification with the cycling condition of denaturation (95 °C, 3 min), anneal (60 °C, 30 s) and extension (72 °C, 5 min), and each sample conducted three repetitions and the specific primers were listed in (Table 1).

## Wound healing and trans-well assay

We purchased the small interfering RNA reagent of si-S100A8 from Sangon Biotech (Shanghai, China) to create the HAP1 cells with *S100A8* silencing, the sequences of si-*S100A8* included the sense (5′-AUGGAAAUUCCCCUUUAUCAG-3′) and anti-sense (5′-GAUAAAGGGGAAUUUCCAUGC-3′), its working concentration and Lipofectamine 3000 (Invitrogen) were applied for cell transfection. The wound healing for cell migration were performed, the 6-well plates (Corning, Corning, NY, USA) containing complete medium and 10% FBS was plated with $4 \times 10^6$ cells. When the cells reached the 90% confluency, a 20 μL pipette tip was used to create a rectilinear scratch and the cells washed by PBS after 48 h incubation and imagined by an inverted microscope (Leica) were performed. For trans-well assay, the 24-well plates containing chamber inserts (8-μm pore) were seeded with $4 \times 10^4$ cells in the upper chamber (300 μL serum-free medium) and the lower chamber was supplemented with 700 μL medium with 10% FBS. Followed by 48 h incubation, the 4% paraformaldehyde and 0.1% crystal violet were used for cell processing, and the cells imaging and counting were conducted by an inverted microscope (Leica).

## Survival analysis

We performed Cox analysis on the TCGA-AML dataset using the R package "survival" to assess the prognostic role of marker genes screened by macrophage mapping and NK/T cells mapping.

## Statistical analysis

The student's test was used to compare the differences in continuous variables between the two groups. All calculations were performed by the R language (version 4.3.1). $p < 0.05$ was considered statistically significant. (*$p < 0.05$, **$p < 0.01$, ***$p < 0.001$, ****$p < 0.0001$)

## RESULTS

### Single-cell mapping of AML

In this study, the AML single-cell dataset was pre-processed by normalization, dimensionality reduction, and clustering, and a total of 3,627 cells were screened for subsequent analysis, and six major cell subpopulations were obtained, including Mast cells, Macrophage cells, NKT cells, blood cells, and B cells, Megakaryocyte (Fig. 1A). Mast cells highly expressed *CTSG, HPGD, IGLL1, SMYD3*; Macrophage cells highly expressed *CD86, MS4A7, CD163*; NKT cells highly expressed *NKG7, GZMA, CCL5, GZMB, GZMH, GZMK, CD3D, CD3G, CD3E, CD2*; Blood cells highly expressed *ALAS2, GYPA, RHCE*; B cells highly expressed *CD79A, MS4A1*; Megakaryocyte highly expressed *ITGA2B, PF4, TUBB1, GP9* (Figs. 1B, 1C). Follow-up analyses revealed the cell subpopulations in each sample and the proportions of each cell subpopulation in different subgroups, and we found that macrophage cells and NK/T cells showed a significant up-regulation in non-CR AML samples (Figs. 1D, 1E). This further reveals that these two cell types may have potentially important regulatory roles in the disease progression of AML.

### Heterogeneity of macrophage cells in AML

In this study, macrophage cells were further clustered and four subpopulations of macrophage cells were obtained: FCAR+ macrophage cells, IL32+ macrophage cells, ELANE+ macrophage cells, FCGR3A+ macrophage cells (Fig. 2A). The expression of *FCAR, IL32, ELANE* and *FCGR3A* is shown in Figs. 2B and 2C. The proportions of each cell subpopulation were next calculated, and the results showed that FCAR+ macrophage cells and FCGR3A+ macrophage cells were elevated in the AML group with non-CR (Fig. 2D). Next, enrichment analysis of biological functions was performed to explore the functions of each cell subpopulation. The results showed that ELANE+ macrophage cells were mainly enriched for processes such as cytoplasmic translation, ribose phosphate metabolic process, ATP metabolic process, oxidative phosphorylation, positive regulation of cell adhesion, electron transport chain, humoral immune response; FCAR+ macrophage cells were mainly enriched for processes such as positive regulation of cytokine production, activation of immune response, positive regulation of MAPK cascade, I-kappaB kinase/NF-kappaB signaling, ERK1 and ERK2 cascade; FCGR3A+ macrophage cells were mainly enriched for processes such as actin filament organization, positive regulation of cytokine production, immune response-regulating signaling pathway; IL32+ macrophage cells were mainly enriched for processes such as cytoplasmic translation, positive regulation of leukocyte activation, positive regulation of cell activation, positive regulation of lymphocyte activation (Fig. 2E). These results suggest that FCAR+ macrophages and FCGR3A+ macrophages may have a key role in chemoresistance and disease progression in AML.

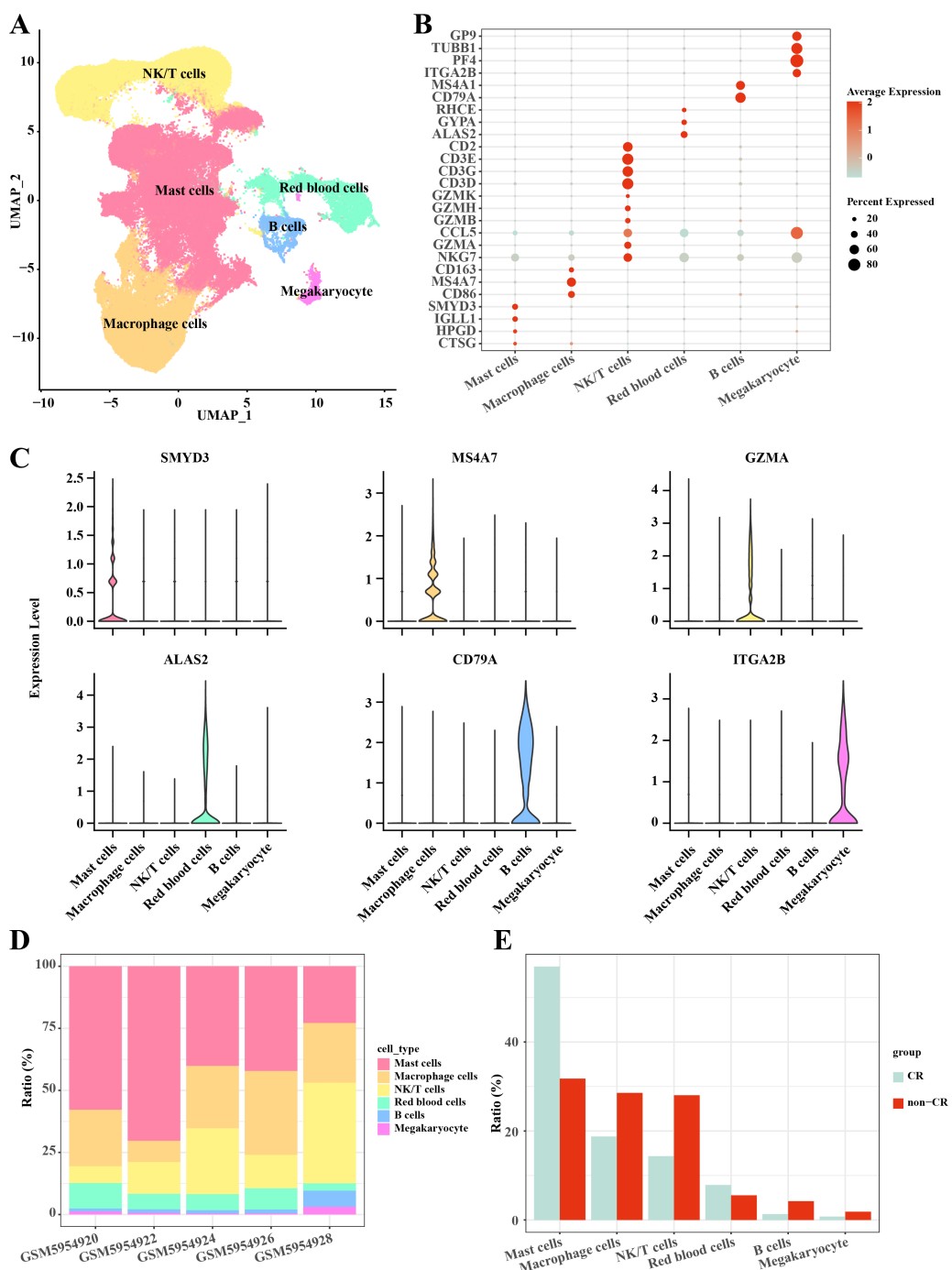

**Figure 1** **Single-cell mapping of AML.** (A) UMAP downscaling plot after AML clustering annotation. (B) Bubble plot of cell subpopulation marker gene expression. (C) Violin plot of cell subpopulation marker gene expression. (D) Proportion of cell subpopulations within AML samples. (E) Difference in cell subpopulation infiltration in CR-AML samples compared to non-CR-AML samples.

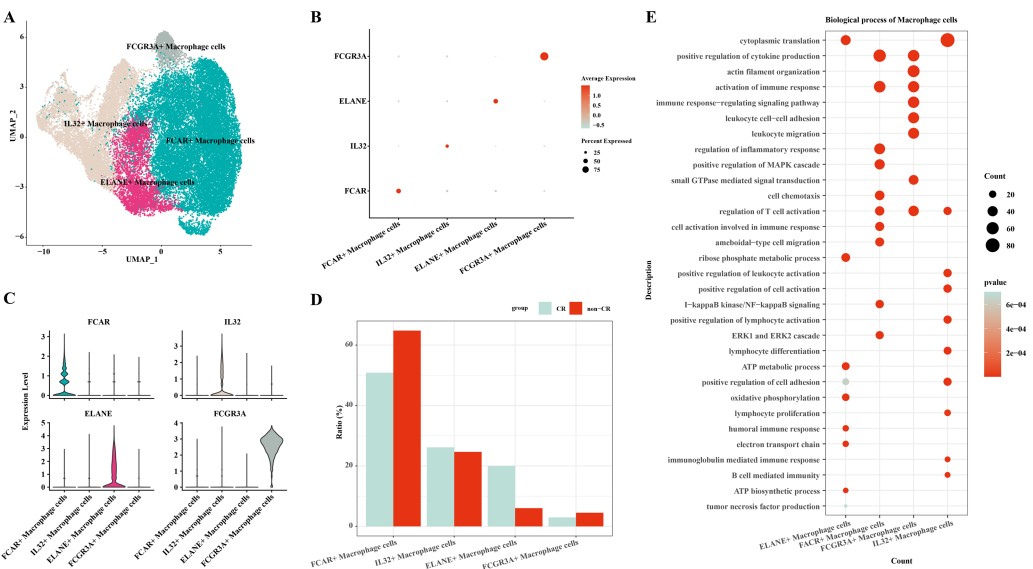

**Figure 2** **Macrophage cells single cell atlas.** (A) UMAP on the infiltration levels of subpopulations of macrophage cells. (B) Bubble plots of the relative expression levels of marker genes in each subpopulations of macrophage cells. (C) Violin plots of the relative expression levels of marker genes in each subpopulation of macrophage cells. (D) Differences in the infiltration ratios of each subpopulation of macrophage cells. (E) Functional enrichment analysis of marker genes in each subpopulation of macrophage cells.

## Differentiation trajectories of macrophage cells

Based on the results of previous analyses, the present study was conducted to analyze the proposed time series of FCAR+ macrophage cells and FCGR3A+ macrophage cells cell subpopulations. The results showed that macrophage cells differentiated into two clusters of cells: cell fate 1 and cell fate 2 (Fig. 3A). The results of the changes of differentiation process genes showed that the genes that were gradually down-regulated were mainly enriched in the processes of protein targeting, ATP metabolic process, and T cell activation, while the genes that were gradually up-regulated were mainly enriched in the processes of regulation of cell cycle phase transition, regulation of mitotic cell cycle phase transition, position regulation of cell cycle, *etc.* (Fig. 3B). Changes in cellular marker genes within each branch showed that genes that were progressively up-regulated were mainly enriched in processes such as T cell activation, oxidative phosphorylation, ERK1 and ERK2 cascade, whereas genes that were progressively down-regulated were enriched in processes such as Ras protein signal transduction and other processes (Fig. 3C). The gene expression results among different cell clusters showed that *FCGR3A* and *BTG2* genes were gradually increased in cell fate2, while *ARHGEF40, FCAR, MYO9B, ATM, PREX1, RAB27A, ROCK1, ATRX, KAT6A, and PRKAG2* genes were progressively up-regulated in the Cell fate1 branch (Fig. 3D).

## Heterogeneity of NK/T cells in AML

In this study, NK/T cells were further clustered into three cell subpopulations: Native CD8+ T cells, natural killer cells and Treg cells (Fig. 4A). Among these cell subpopulations, native CD8+ T cells up-regulated the expression of *TRABD2A, MAL, FAM102A, SH3YL1*;
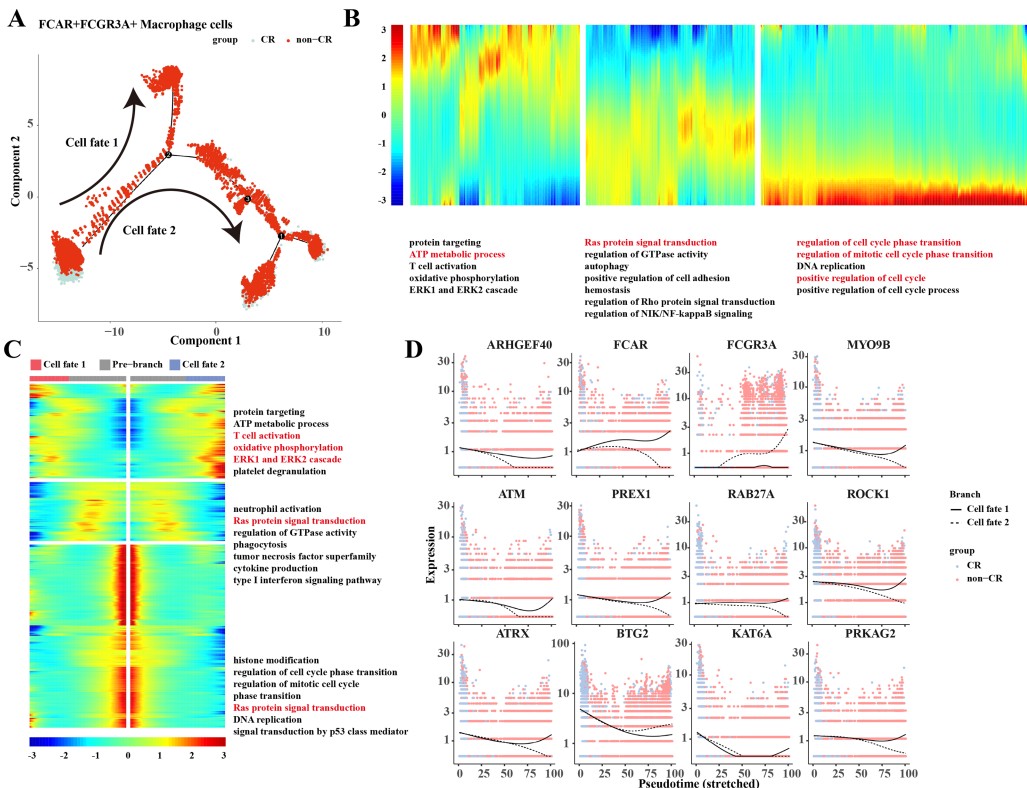

**Figure 3 Pseudotemporal analysis of Macrophage cells.** (A) Macrophage pseudotime scatter plot. (B) Macrophage pseudotime analysis heatmap. (C) Macrophage pseudotime analysis branching heatmap. (D) Expression of key genes in cell branching with pseudotime.

Natural killer cells highly expressed *CLIC3, TRDC, KLRG1*; Treg cells up-regulated *MBOAT7, SOX4, CAPG, GRN, CD82, HLA-DMA, HLA-DMA, HLA-DMA, HLA-DMA, HLA-DMA*, and *HLA-DMA*, *CD82, HLA-DMA* (Figs. 4B, 4C). The results of the percentage of infiltration of each cell subpopulation showed that Natural killer cells were elevated in the non-CR AML group compared to the CR AML group (Fig. 4D). This implies that NK cells may also play an important regulatory role in the development and progression of AML.

## Differentiation trajectories of natural killer cells

We performed functional enrichment analysis of highly expressed genes in natural killer cells, and the results showed that the highly expressed genes in natural killer cells were mainly enriched in negatively regulating immune responses (Fig. 5A), which suggests that natural killer cells may be dysfunctional or have developed chemotherapy tolerance. To explore this phenomenon, we performed a mimetic chronological analysis of natural killer cells (Fig. 5B). We further investigated the genes that play key roles in cell evolution, and the results showed that genes such as *HBB, HBA1, CXCR4, RPS3A* and *VIM* were expressed at a higher level in the CR group, while genes such as *RPS26, IFITM1, IER2, ID3, S100A8* and *S100A9* were expressed at a higher level in the non-CR group (Figs. 5C, 5D).

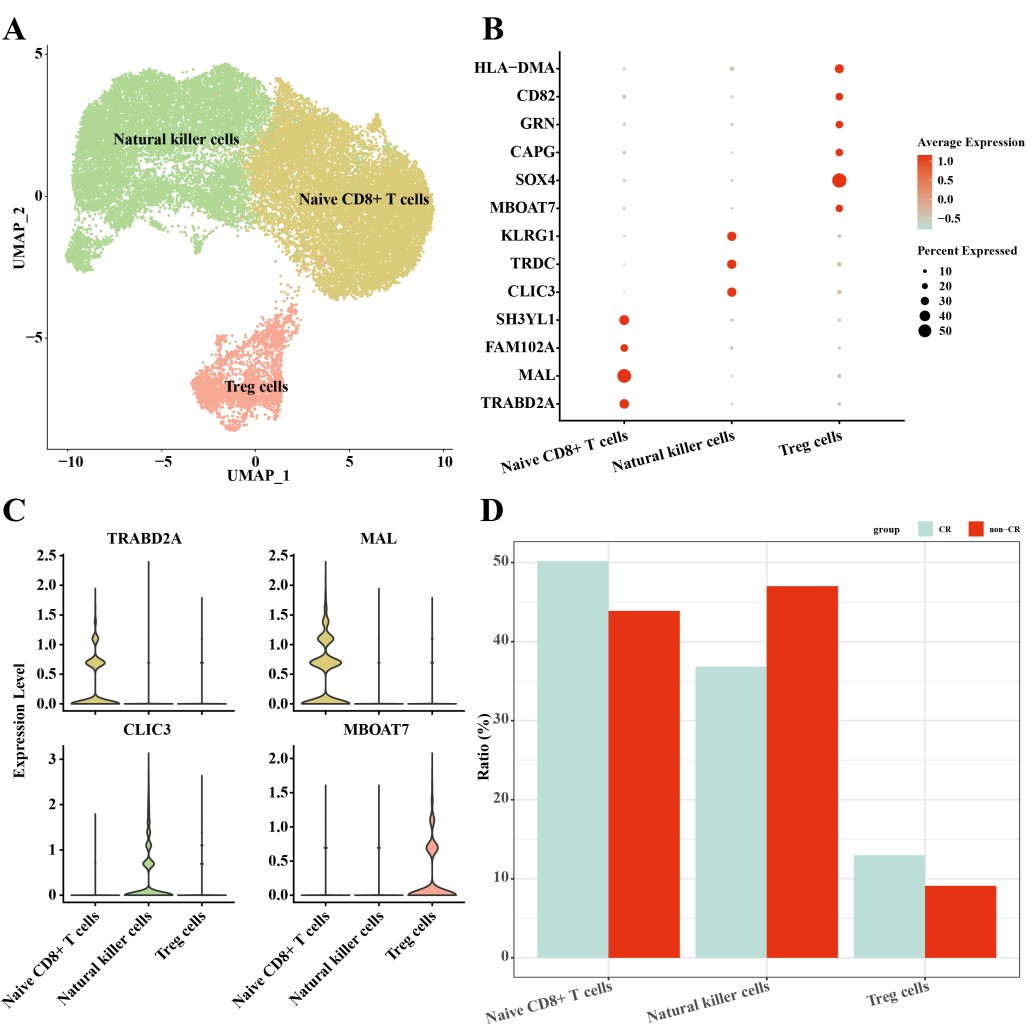

**Figure 4   Single-cell mapping of NK/T cells.** (A) UMAP plots of NK/T cells. (B) Bubble plots of marker genes highly expressed in each subpopulation of NK/T cells cells. (C) Violin plots of marker genes highly expressed in each subpopulation of NK/T cells cells. (D) Relative infiltration ratios of each cell subpopulation of NK/T cells in non-CR-AML and CR-AML.

It is worth mentioning that *HBA1, S100A8* and *S100A9* play important roles in mediating drug resistance in cancer cells, which also provides insights into the mechanism of drug resistance in AML.

## Interactions between macrophage cells and natural killer cells

Macrophage cells usually function as antigen presenters to activate T cells or B cells when an immune response occurs in the body. Therefore, we investigated the cell communication process between macrophage cells and natural killer cells. The results showed that NK cells and macrophage cells interacted *via* HLA-E-KLRK1, HLA-E-KLRC1, HLA-E-CD94:NKG2A, CLEC2B-KLRB1 ligand–receptor pairs during Cell-Cell Contact communication (Fig. 6A). The ligand–receptor pairs that play a key role in the regulation

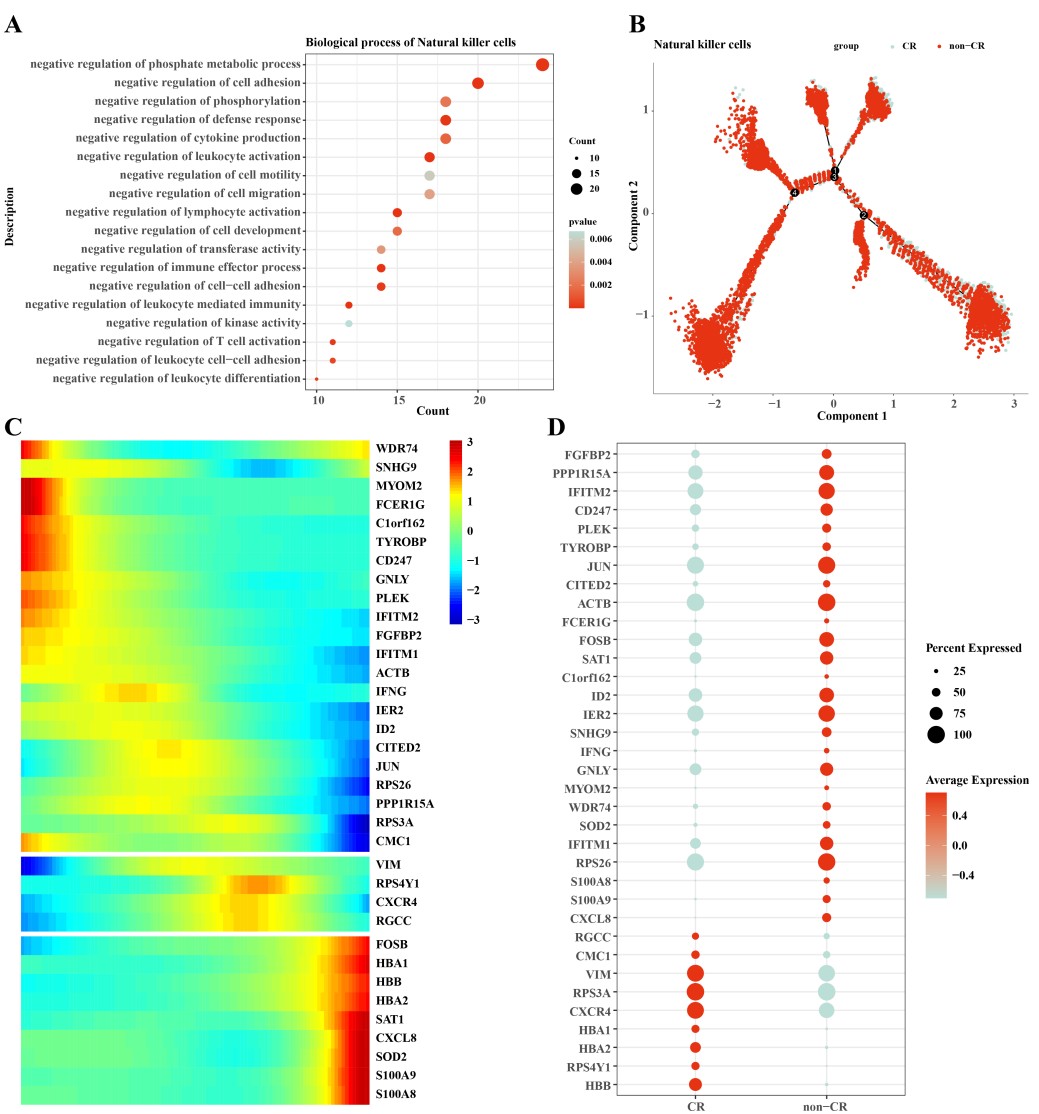

**Figure 5  Pseudotemporal analysis of NK/T cells.** (A) Biofunctional enrichment analysis of highly expressed marker genes in NK/T cells. (B) NK/T cell pseudo-temporal analysis of branching scatter plots. (C) NK/T cell pseudo-temporal analysis of branching heatmaps. (D) Difference in expression of key marker genes in NK/T cells in non-CR-AML and CR-AML.

of secreted signaling mainly include LGALS9-CD45, CCL3L1-CCR1, CCL3-CCR1 (Fig. 6B).

## Immunoreactivity and prognostic analysis of macrophage cells and natural killer cells in AML

To verify the activity of macrophages and NK cells, we performed AUCell analysis, and in NK cells activation, there was higher activity in the non-CR group, while in NK cells proliferation, there was no significant difference between the CR and non-CR groups in NK/T cells activity was not significantly different (Figs. 7A, 7B). In macrophage activation,

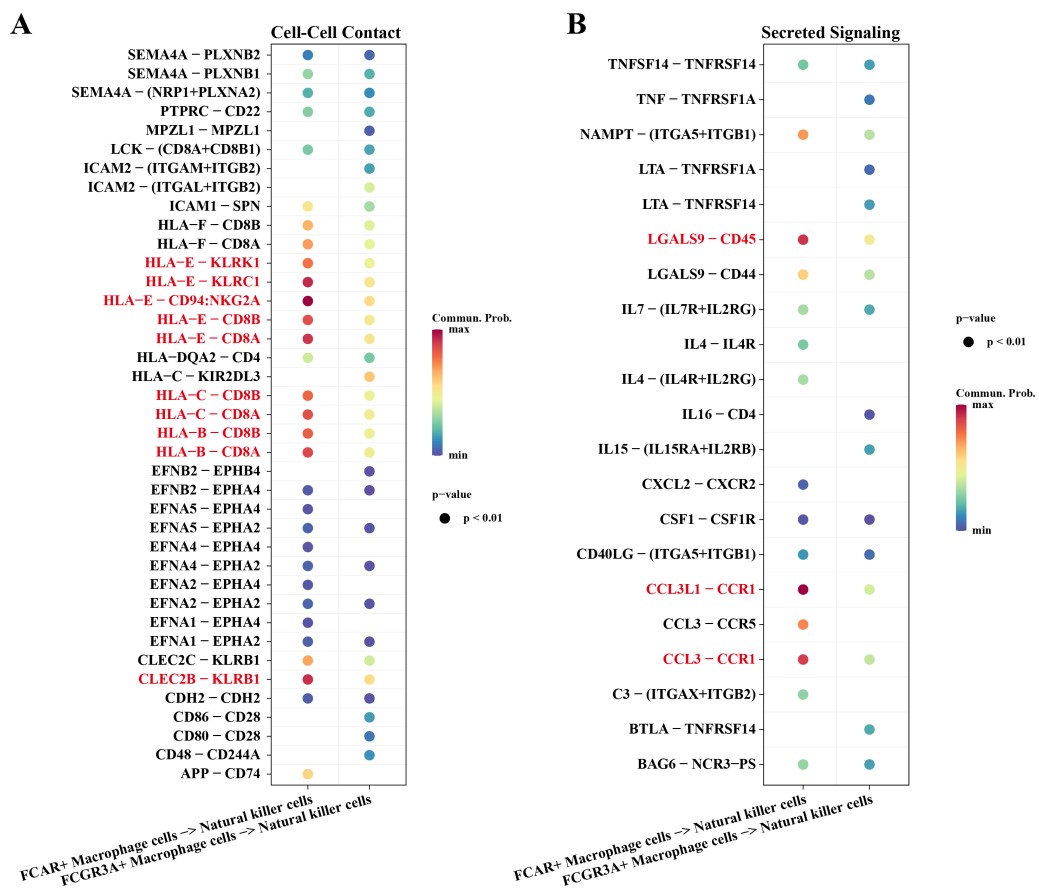

**Figure 6** **Cellular communication analysis.** (A) Pairing-receptor pairs that mediate contact between macrophages and natural killer cells. (B) Pairing-receptor pairs that mediate secreted signaling between macrophages and natural killer cells.

FCG3A+ macrophage cells and FCAR+ macrophage cells had higher activities, while in macrophage migration, FCAR+ macrophage cells had higher activities (Figs. 7C, 7D). These results suggest that NK cells and FCAR+ macrophage cells may be involved in non-CR in chemotherapy-resistant organisms in AML. We next performed survival analysis, which showed that patients in the AML group with high expression of HBA1 had a better prognosis, whereas patients in the AML group with high expression of *PREX1, S100A8* and *S100A9* had a worse prognosis (Figs. 8A–8D). The expression profiles of these genes may serve as potential biomarkers of AML prognosis and inform the individualized treatment of AML.

## Upregulation of marker genes in leukemia cells affected AML progression

Finally, we detected the expression of marker genes of macrophage and NK cells, and found that five genes of *FCAR, FCGR3A, PREX1, S100A8* and *S100A9* all were significantly overexpressed in the leukemia cells of HAP1 compared with that in the normal stroma cells of HS-5 ($p < 0.05$, Fig. 9A). The wound healing assay revealed that the wound closure

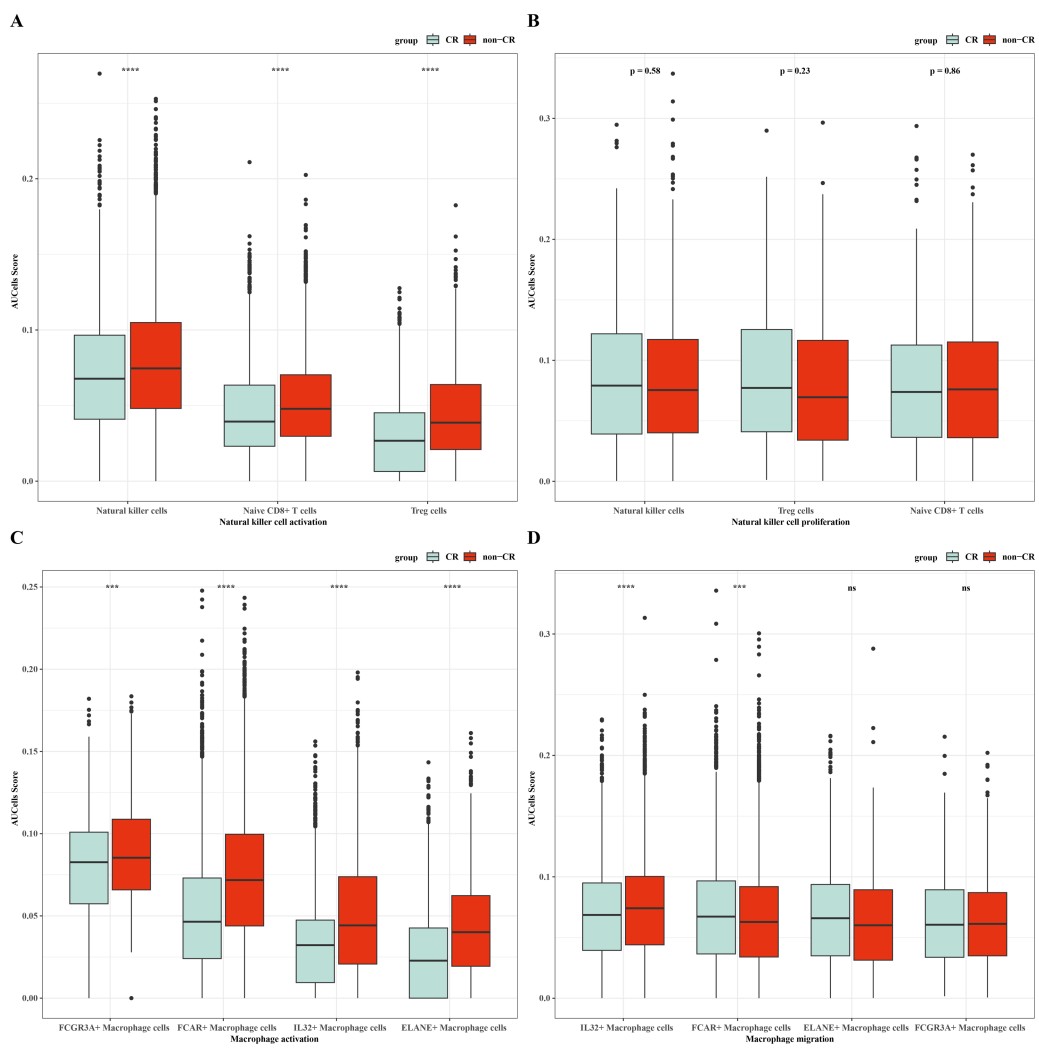

**Figure 7   NK/T cells and macrophage cells activity were analyzed.** (A) AUC cell analysis of NK/T cells cell subsets in nature killer cells activation. (B) AUC cell analysis of NK/T cell subsets in nature killer cells proliferation. (C) AUC cell analysis of macrophage cell subsets in macrophage activation. (D) AUC cell analysis of macrophage cell subsets in macrophage activation.

rate of HAP1 cells with si-*S100A8* silencing was significant reduced ($p < 0.05$, Figs. 9B, 9C) and the cell numbers of HAP1 cells with si-*S100A8* silencing in the trans-well assay was also significantly decreased ($p < 0.05$, Figs. 9D, 9E), indicating the *S100A8* may be involved in the leukemia migration and invasion.

## DISCUSSION

The treatment of AML is based on chemotherapy, but given the heterogeneity of the disease, clinically targeted chemotherapy regimens often face the challenge of leukemic cell resistance (*Ediriwickrema, Gentles & Majeti, 2023*). Immune cells are potential regulators of AML progression and play a role in the regulation of AML chemoresistance (*Mohamed*

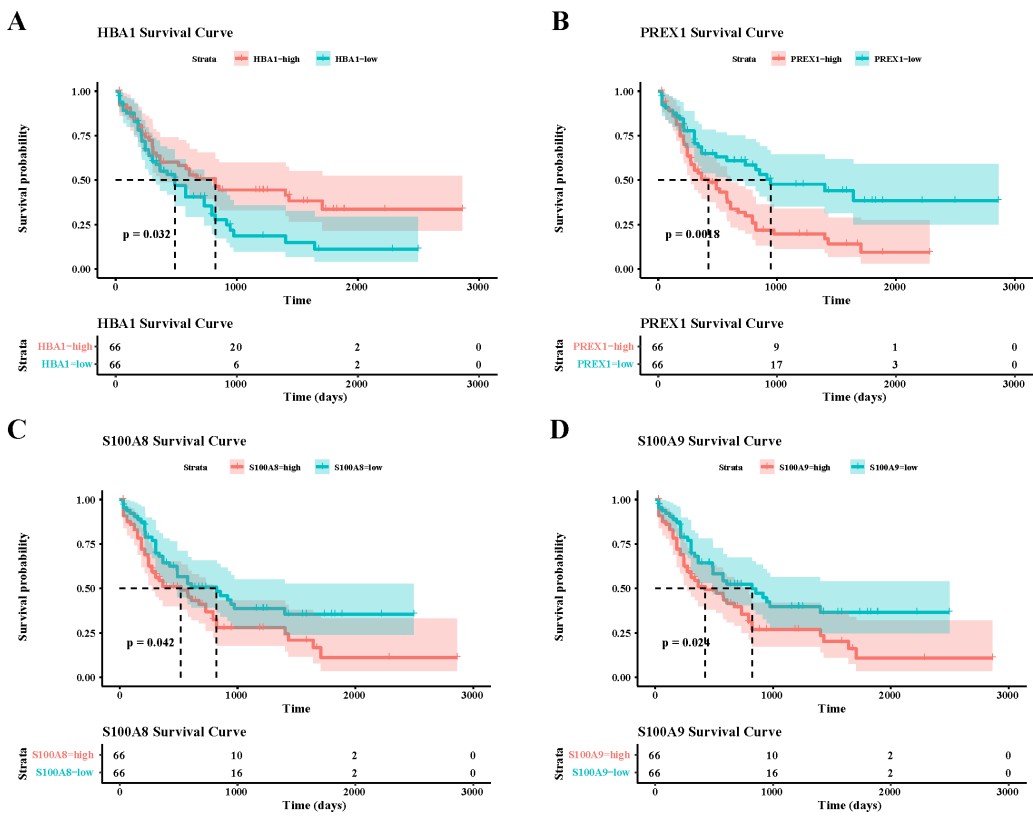

**Figure 8  Survival analysis of AML by key genes mediating AML drug resistance.** (A) Survival analysis of HBA1 in AML. (B) Survival analysis of PREX1 in AML. (C) Survival analysis of S100A8 in AML. (D) Survival analysis of S100A9 in AML.

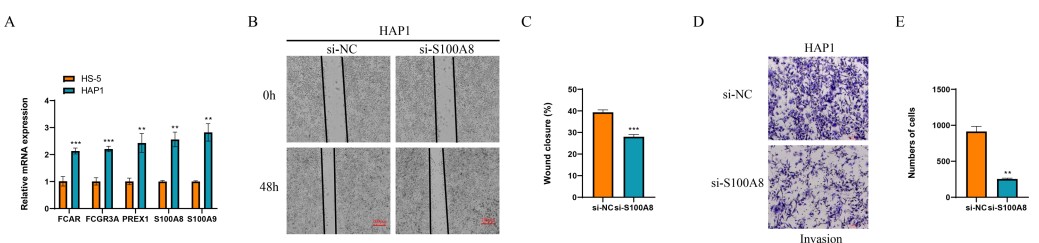

**Figure 9  Functional verification of marker genes *in vitro*.** (A) qPCR for the gene expression in cells. (B–C) Wound healing assay for cell migration. (D–E) Trans-well assay for cell invasion.

*Jiffry et al., 2023*). Thus, this study focuses on the revelation of immune cell profiles in AML single-cell transcriptomic data and elucidation of key immune cells and their marker genes affecting AML chemoresistance to elucidate the regulatory mechanisms of key marker genes of immune cells on AML chemoresistance with functional enrichment, differentiation trajectory construction and prognostic analysis.

In this study, single-cell transcriptomic analysis revealed that Macrophage cells and NK/T cells showed significant up-regulation in non-CR-AML samples relative to CR-AML

samples, which also implies that these two types of cells play a role in the regulation of chemotherapeutic resistance in AML. Macrophages play a tangible regulatory role in AML progression, *e.g.*, M2 polarization of macrophages synergizes with the inhibition of lipid peroxidation in AML, which promotes the progression of this disease (*Liu et al., 2022*). In contrast, M2 macrophages produce a variety of cytokines, chemokines and peptide growth factors to remodel the microenvironment and mediate the release of histone B to enhance chemotherapeutic drug resistance in cancer cells (*Shree et al., 2011*; *Bruchard et al., 2013*). In contrast, M1 macrophages mediate cancer cell drug sensitization by inhibiting bone marrow cell recruitment and activating immune cell killing activity (*Wang et al., 2024*). NK cell-based overt therapy is a novel therapeutic option for palliation of hematologic malignancies, given the targeted and persistent characteristics of NK cells in the tumor microenvironment (*Merino, Maakaron & Bachanova, 2023*; *Meng et al., 2024*). Binding of the NKG2D receptor on the surface of this cell line to the MICA/B ligand on the surface of AML cells promotes NK cell activation and AML cell killing, thus indirectly affecting the drug sensitivity of AML cell lines (*Xing & Ferrari de Andrade, 2020*; *Baragano Raneros et al., 2015*).

Given the important role of macrophages in the regulation of AML drug resistance, this study revealed an elevated proportion of FCAR+ macrophage cells and FCGR3A+ macrophage cells in the non-CR AML group by further clustering. FCAR encodes an immunoglobulin that is widely expressed on the surface of neutrophils, monocytes, and macrophages, among other immune cell surfaces, mediating anticancer phagocytosis and stimulation of inflammatory mediators. This gene affects AML progression by activating the anti-cancer immune response in macrophages (*Ke et al., 2024*; *Kelley et al., 2011*; *Shahrajabian & Sun, 2023*). FCGR3A mediates antibody-dependent cytotoxicity, and thus overexpression of this gene on the surface of immune cells mediates the release of cytotoxins and cytokines to kill target cells, and it is its immune activation that makes it potentially valuable for immunotherapy in AML (*Li et al., 2022*; *De Taeye et al., 2020*). The present study reveals that the genes gradually down-regulated during the differentiation of Macrophage cells are enriched in the ATP metabolic process, T cell activation and other processes related to the regulation of cellular energy metabolism and immune cell activity, while the genes gradually up-regulated are enriched in the pathways related to the regulation of the cell cycle. Both cellular energy metabolism and cell cycle dynamics are inextricably linked to the mediation of drug resistance in cancer cells. For example, the polarization state of macrophages is regulated by the level of glycolysis, and metabolites such as pyruvate and lactate released by cells *via* glycolytic metabolism directly affect the chemotherapeutic resistance of cells (*Yuan et al., 2022*; *Li et al., 2023*). Regarding cell cycle regulation, inactivation of cell cycle inhibitors such as p53, p21 and p27 released by cells in G1/S and G2/M phases can allow cancer cells to continue dividing in the presence of DNA damage or replication errors, a phenomenon that can lead to resistance of cancer cells to DNA-damaging drugs (*Clay & Fox, 2021*).

The results of the infiltration ratio of each subpopulation of NK/T cells showed that the percentage of NK cells was significantly higher in the AML group with non-CR compared to the AML group with CR, and the NK cell marker genes that were differentially expressed

between the CR and non-CR groups mainly included *HBA1, PREX1, S100A8* and *S100A9*. The *HBA1* gene is mainly involved in the regulation of hematopoiesis, promotes the differentiation and maturation of erythrocytes, and affects the synthesis and stability of hemoglobin, which directly regulates AML progression (*Luo et al., 2022*). *PREX1* gene is able to regulate the process of cell cycle, and can affect the function of cell cycle checkpoints, and is involved in the process of apoptosis, and repair of damaged DNA (*Liu et al., 2016*; *Stadler & Richly, 2017*). The repair of cellular DNA damage can regulate the development of chemoresistance in cancer cells, and has a practical role in the regulation of drug resistance in AML patients (*Salehan & Morse, 2013*). The *S100A8* and *S100A9* genes, on the other hand, are mainly involved in the process of immune response, and they can regulate the function of T and B cells in the bone marrow microenvironment of AML patients, affecting the production of antibodies and the process of antigen presentation (*Mondet, Chevalier & Mossuz, 2021*). Notably, the *S100A8* and *S100A9* genes mediate oxidative stress, activation of apoptotic pathways, and promotion of cellular autophagy in AML cells to affect survival and sensitivity to chemotherapy in AML patients (*Mondet, Chevalier & Mossuz, 2021*; *Bottcher et al., 2022*). In *vitro* assay, we also observed that these genes were upregulated in the leukemia cells affected AML progression. Taken together, the NK cell gene marker genes revealed in this study may be important factors in the regulation of AML chemosensitivity.

In this study, the interaction mechanism between NK cells and Macrophage cells was mined by cell communication analysis, and the results showed that NK cells and Macrophage cells interacted through ligand–receptor pairs such as HLA-E-KLRK1, HLA-E-KLRC1, and HLA-E-CD94:NKG2A. Among them, HLA-E is a member of the non-classical human leukocyte antigen family that modulates the killing activity of NK cells and regulates the intensity and duration of the immune response by binding to MHC-I-like molecules and transmitting inhibitory signals to immune cells such as NK cells, macrophages and CD8+ T cells (*Ravindranath et al., 2019*; *He et al., 2023*). Meanwhile, the highly expression of HLA-E as a ligand for KLRC1, KLRK1, and NKG2A/CD94 on the surface of NK cells can help cancer cells evade immune killing by NK cells (*Liu et al., 2023*; *Mac Donald et al., 2023*). In Secreted Signaling, NK cells and macrophage cells play a major role with ligand–receptor pairs such as LGALS9-CD45, CCL3-CCR1, *etc.* LGALS9, a member of the galactoglucan lectin family and an emerging target for cancer immunotherapy, exerts anti-tumor immune responses by regulating the homeostasis of the immune microenvironment and Tim-3 signaling to exert anti-tumor immune responses (*Lv et al., 2023*). CD45 targeted by LGALS9 has been demonstrated by animal experiments to have a role in promoting cell stemness and resistance to radiotherapy in cancer cells, and a related study revealed that up-regulated expression of CD45 promotes cancer cell survival in mice (*Park et al., 2021*). This has implications for the present study on the mechanism of drug resistance in AML, *i.e.,* LGALS9 targeting and inhibiting CD45 affects AML chemoresistance through the regulation of immune cell activity in the bone marrow microenvironment. These results suggest that there may be immune dysregulation in AML patients who develop chemotherapy resistance during AML receiving chemotherapy, leading to non-CR.

In summary, we analyzed single-cell transcriptomic data of AML to reveal the potential regulatory role of NK cells and macrophages in AML chemoresistance. Functional enrichment analysis further elucidated the regulatory functions of the two immune cells in the regulation of AML chemoresistance. Pseudotime analysis combined with cellular communication analysis revealed the differentiation process of chemotherapy-resistant NK cells and macrophages as well as the ligand–receptor pairs of their interactions. The revelation of the molecular mechanisms of immune cells in AML in this study is expected to provide guidance for alleviating chemotherapy resistance in AML. However, limitations of this study remain. The data in this study were mainly derived from public databases and lacked certain experimental samples and cellular experimental data. Follow-up cell or tissue sample experiments are needed to further validate the regulatory mechanisms of immune cell marker genes in AML on disease chemoresistance.

**Abbreviations**

| | |
|---|---|
| **AML** | Acute myeloid leukemia |
| **DEGs** | Differentially expressed genes |
| **GEO** | GENE EXPRESSION OMNIBUS |
| **TCGA** | The Cancer Genome Atlas |
| **CR** | Complete remission |
| **KEGG** | Kyoto Encyclopedia of Genes and Genomes |
| **GO** | Gene Ontology |
| **scRNA-seq** | Single-cell cell sequencing |

### Funding

The study was supported by the Natural Science Basic Research Program of Shaanxi (2020JQ-460). The funders had no role in study design, data collection and analysis, decision to publish, or preparation of the manuscript.

### Grant Disclosures

The following grant information was disclosed by the authors:
Natural Science Basic Research Program of Shaanxi: 2020JQ-460.

### Competing Interests

The authors declare there are no competing interests.

### Author Contributions

- Jing Gao conceived and designed the experiments, analyzed the data, prepared figures and/or tables, authored or reviewed drafts of the article, and approved the final draft.
- Xueqian Yan conceived and designed the experiments, analyzed the data, authored or reviewed drafts of the article, and approved the final draft.
- Dan Fan performed the experiments, analyzed the data, authored or reviewed drafts of the article, and approved the final draft.
- Yuanchun Li performed the experiments, analyzed the data, prepared figures and/or tables, and approved the final draft.

## Data Availability

The datasets generated and/or analyzed are available at NCBI: GSE198681.

The raw data is available at GitHub, Zenodo and Figshare:

– https://github.com/Shmily98306/All-raw-data.git

- Shmily98306. (2024). Shmily98306/All-raw-data: updated raw data (v.1.1.1). Zenodo. https://doi.org/10.5281/zenodo.13756036

- Gao, Jing; Yan, Xueqian; Fan, Dan; Li, Yuanchun (2024). 0_Raw_data.zip. figshare. Dataset. https://doi.org/10.6084/m9.figshare.26891560.v1.

The raw data is available in the Supplemental Files.

## Supplemental Information

Supplemental information for this article can be found online at http://dx.doi.org/10.7717/peerj.18521#supplemental-information.

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
