# Peer review of "Single-cell data revealed the function of natural killer cells and macrophage cells in chemotherapy tolerance in acute myeloid leukemia"

_PeerJ, doi:10.7717/peerj.18521_

## Round 0.1 · original submission · Minor Revisions

After careful consideration of the reviewers' comments, I am pleased to inform you that your manuscript is potentially acceptable for publication, pending minor revisions. Both reviewers have provided valuable feedback and have suggested some minor changes to improve your manuscript. Please address these comments carefully and submit a revised version of your manuscript along with a point-by-point response to the reviewers' comments

Reviewer 1 ·

Basic reporting

This manuscript discusses the role of NK cells and macrophages in tolerance to chemotherapy for acute myeloid leukemia. Particularly, it focuses on depicting the phenomena related to immune cells regulating drug resistance.
The manuscript is reasonably well written, generally well cited, and methods description is fairly detailed. The findings are well supported by the presented data. However, in some instances, it is hard to grasp exactly what is aimed to be conveyed by some of the graphs.

There are some minor concerns that should be addressed before publication, related to the clarify of the presented data, as well as some stylistic issues:

- Fig 1B Bubble plot lacks average expression units, or if dimensionless, the expression that this was calculated with.
- Fig 1D feels arbitrary. Instead, it would be insightful to plot the mean distribution of patients with CR and non-CR, which would complement Fig 1E well
- Expression level in plots likely refers to logarithms, but it’s encouraged to reveal the formula
- Line 201: response
- Caption of figure 2 is repeated (D&E)
- In general, many plots and color scales lack units, and makes plots hard to follow and interpret

Experimental design

Experimental result is robust.

Validity of the findings

No comment

Reviewer 2 ·

Basic reporting

no comment

Experimental design

no comment

Validity of the findings

no comment

Additional comments

In this study, the author revealed the chemotherapy tolerance of natural killer cells and macrophage cells in acute myeloid leukemia by using the single-cell RNA-seq analysis. The experimental design is reasonable and the arguments are sufficient, but there are still some problems that the author needs to improve before the publication of this paper.
1. Line 32, the Macrophage cells subpopulations were upregulated in non-CR AML. Usually, the “upregulation” is used to characterize gene expression. Please consider whether it is appropriate to “regulated” in this sentence. Line 34, the first letter “differentially” of the sentence should be capitalized.
2. Line 33, What is the evidence supporting the involvement of macrophage in resistance to acute myeloid leukemia disease. The increased cell proportion of macrophage alone does not supports this title.
3. Line 25, AUCell enrichment scores were used to assess the immunoregulatory functions of immune cells, and pseudotime analysis for differentiation trajectories, the results are lacking.
4. Line 44, Please briefly describe the main arguments and the significance of this study in the conclusion section.
5. Line 58-59, Please unify the tenses of this sentence. Please check the grammar and tenses, the confusing grammar may hinder the smooth publication of this article.
6. Line 70, the immune escape of AML cells is an important factor in disease recurrence. What are the factors that promote immune escape of cells.
7. In the results, can the author follow each result with a short summary or hypothesis.
8. Line 240-246, the exact function of HBB, HBA1, CXCR4, RPS3A and VIM is what, How do they affect drug resistance in cells.
9. Line 287-291, the M2 polarization of macrophages affected the progression of disease. Did the authors confirm that macrophages in this paper also undergo macrophage M2 polarization and participate in cell drug resistance.
10. What are the important prospects of this article. How can the results be used to guide future research.

---

## Round 0.2 · accepted · Accept

After careful review of your revision and the positive recommendations from both reviewers, I am pleased to inform you that your manuscript has been accepted for publication. Please proceed with the preparation of your production materials according to the journal's requirements. Our production team will be in contact with you shortly regarding the next steps in the publication process.

Reviewer 1 ·

Basic reporting

The authors have carefully and thoroughly addressed my comments, by response in the document, as well as changes in the manuscript.

Experimental design

Previous review was satisfactory.

Validity of the findings

Previous review was satisfactory.

Reviewer 2 ·

Basic reporting

no comment

Experimental design

no comment

Validity of the findings

no comment

Additional comments

In this study, the authors revealed the chemotherapy tolerance of natural killer cells and macrophages in acute myeloid leukemia through single-cell RNA seq analysis. The experimental design is reasonable and the argument is sufficient. The author has responded point-to-point to the reviewer's comments and revised the manuscript. It now appears to be complete, and I do not have any new comments